# Laser Attenuation and Ranging Correction in the Coal Dust Environment Based on Mie Theory and Phase Ranging Principle

**Ben Li [1], Shanjun Mao [1,\*] and Hong Zhang [2]**

[1] Institute of Remote Sensing and Geographic Information System, Peking University, Beijing 100871, China; benli@pku.edu.cn

[2] Beijing Longruan Technologies Co., Ltd., Beijing 100871, China; zhanghong@longruan.com

\* Correspondence: sjmao_@pku.edu.cn

**Abstract:** The inadequate ventilation and complex environments in underground coal mines lead to a high concentration of dust particles. As a result, the health of the miners and the accuracy of laser rangefinder measurements are endangered. It is crucial to enhance the laser rangefinder's efficiency to mitigate health risks and reduce labor intensity. In this study, we propose a laser power attenuation model and a ranging correction model to address the issues of laser power attenuation and inaccurate ranging in coal dust environments. The proposed models are based on theoretical analysis and practical experiments, and both are dependent on the dust particle size (<250 μm) and mass concentration. Firstly, we assessed the factors that caused laser power attenuation and demonstrated that our proposed model could accurately predict them (maximum residual of 0.06). Secondly, we obtained the connection between the attenuation coefficient and dust concentration by applying the Lambert–Beer law. Lastly, we established the ranging correction model by collecting laser wavelength information. The outcomes show that the root mean square error of the corrected values ranges between 0.27 and 0.47 mm. To summarize, our suggested model and correction technique can efficiently enhance the precision of laser rangefinder measurements, thus improving underground work in coal mines.

**Keywords:** coal dust control; laser attenuation; dust ranging correction; Mie theory; phase ranging

## 1. Introduction

Advanced digital technologies, such as artificial intelligence, are gradually transforming the mining industry. The intelligent system, by accurately monitoring equipment location on fully mechanized mining faces in real-time, can effectively adjust their pose by incorporating the geological model. During the coal mining process, the laser rangefinder plays an important role as a high-precision distance measurement system [1–3]. Unfortunately, due to poor ventilation and complex working environments in coal mines, a high concentration of dust particles inevitably absorbs and scatters the laser signal transmitted by the instruments [4]. This absorption can cause a loss of ranging and orientation accuracy or even system failure. Existing reports on coal dust treatment and prevention mainly focus on studying the diffusion characteristics of dust and enhancing dust removal efficiency using numerical simulations and pragmatic experiments [5–10]. However, the diffusion of coal dust is unavoidable in crucial regions such as roads and mining faces, affecting the performance of laser instruments severely.

To enhance the accuracy of laser rangefinders in coal dust environments, one must initially examine power attenuation characteristics to obtain cut-off parameters for ranging failure. Additionally, it is necessary to study ranging correction methods that fall within the workable parameter range. Several studies have analyzed the laser attenuation characteristics in the presence of sand and dust particles using various methods. For example,

based on Mie theory, Wu et al. [11] examined the laser attenuation characteristics for the single scattering of sand and dust particles with a specific size distribution. Wang et al. [12] simulated the impact of atmospheric visibility on laser intensity under the single scattering and multiple scattering of sand and dust particles. Yang et al. [13] proposed a laser attenuation model to examine the relationship between sand/dust particle size and visibility. Grabner et al. [14] proposed a model for the relationship between visibility and extinction coefficient under different effective particle radii. Huang et al. [15] established and verified an empirical prediction model for laser links in a droplet environment.

Numerical simulation is also an effective method to study laser attenuation characteristics. Liu et al. [16] used the ray tracing algorithm to simulate the scattering characteristics of raindrops in visible and infrared bands. Guo et al. [17] used Monte Carlo simulation to examine the laser pulse shape and energy attenuation in rain conditions. Zhang et al. [18] used the ray tracing algorithm to simulate the attenuation of a laser passing through a titanium powder cloud. In addition, some researchers have examined the correlation between laser attenuation and other parameters, such as the effective number of particles [19], powder cloud shadow [20], laser level signal [21], wavelength [22], and visibility [23–25].

In addition, the error of laser ranging in a particle environment has received considerable research attention. For example, Ryde et al. [26] evaluated the reliability and accuracy of a laser scanner under rain and dust conditions. Nasyrov et al. [27] experimentally studied the relationship between the optical density of dust clouds and the ranging contrast of objects. Pang et al. [28] proposed dual-wavelength frequency-modulated continuous wave (FMCW) laser detection technology to counter the interference of aerosol particles on ranging. Xu et al. [29] examined the effect of dust concentration and relative humidity on the laser ranging error and established relevant error correction formulas for high humidity, dusty, and complex environments. Hou et al. [30] established a back propagation (BP) neural network model to predict the ranging error of laser radar based on the ranging results for a mine goaf in a dusty environment.

Despite a large amount of related work, few laser attenuation models and ranging error correction methods have been reported, particularly in dusty environments. Moreover, most of the research heavily relies on empirical methods and does not integrate theories and experimentation. In this study, we aim to fill these gaps by proposing models for power attenuation and ranging error that leverage Mie theory and the phase ranging principle. We conducted experiments to capture the laser power data and distance measurements of the laser rangefinder under different sizes and concentrations of dust particles. Our proposed exponential attenuation model can accurately fit the observed data, and we compared its statistical characteristics with the other model. We estimated the cut-off distances for ranging based on the Lambert–Beer law and our proposed coefficient model. In addition, we calculated the correction values of measurements using our proposed ranging correction model, which can be used to eliminate the true ranging error. Finally, we verified the efficacy of the correction model by estimating the correction accuracy.

This article is structured as follows. Section 2 presents the laser attenuation theories, from which we extract vital modeling factors, i.e., the particle size and the mass concentration. Section 3 outlines the settings of the conducted experiments, encompassing the environment and materials. In Section 4, we demonstrate and discuss the results of the experiments. We model the laser attenuation, cut-off distance, and the ranging errors in particle size and mass concentration variables. Finally, conclusions and summaries are made in Section 5.

## 2. Theoretical Modeling

### 2.1. Laser Attenuation Theory

According to Maxwell's electromagnetic theory, the electric field of a one-dimensional planar laser pulse propagating in a vacuum can be expressed as follows:

$$\vec{E}(x) = \vec{E_0}(x) \exp(i(kx - \omega t)), \tag{1}$$

where $x$ denotes the propagation position, $k$ denotes the wave vector, $\omega$ denotes the angular frequency, and $t$ denotes the propagation time. The propagation of a laser is affected by the reflection, refraction, and scattering in the medium, causing a change in the optical path as well as energy transfer. When the laser beam is propagating in a medium with a refractive index of $n(\lambda) = n_1(\lambda) + in_2(\lambda)$, the wave vector is $nk$, so the electric field can be expressed as:

$$\vec{E}(x) = \vec{E}_0(x)\exp(-n_2 kx)\exp(i(kn_1 x - \omega t)), \tag{2}$$

It can be seen from Equation (2) that the imaginary part of the refractive index of the medium determines the attenuation of laser intensity, while the real part governs the variation in the propagation velocity. Therefore, the medium has a significant influence on laser propagation. The Mie theory provides a rigorous solution to Maxwell's equations that describes the propagation of electromagnetic waves in a medium with uniform spherical particles or scatterers, and it is suitable for any particle size. In the coal dust environment, the incident wavelength is denoted as $\lambda$, the dust particle size is $r$, the refractive index is $n$, and the dimensionless size parameter is $\alpha = \frac{2\pi}{\lambda}r$. According to the Mie theory, the extinction cross section $C_{ext}$ and the extinction coefficient $Q_{ext}$ are expressed as follows:

$$\begin{aligned} C_{ext} &= \frac{2\pi r^2}{\alpha^2}\sum_{i=0}^{\infty}(2i+1)\mathrm{Re}(a_i + b_i), \\ Q_{ext} &= \frac{2}{\alpha^2}\sum_{i=0}^{\infty}(2i+1)\mathrm{Re}(a_i + b_i), \end{aligned} \tag{3}$$

In Equation (3), $a_i$ and $b_i$ denote the Mie scattering coefficients, which are expressed as:

$$\begin{aligned} a_i &= \frac{\psi_i'(n\alpha)\psi_i(\alpha) - n\psi_i'(\alpha)\psi_i(n\alpha)}{\psi_i'(n\alpha)\zeta_i(\alpha) - n\zeta_i'(\alpha)\psi_i(n\alpha)}, \\ b_i &= \frac{n\psi_i'(n\alpha)\psi_i(\alpha) - \psi_i'(\alpha)\psi_i(n\alpha)}{n\psi_i'(n\alpha)\zeta_i(\alpha) - \zeta_i'(\alpha)\psi_i(n\alpha)}, \end{aligned} \tag{4}$$

where $\psi_i$, $\zeta_i$, $\psi_i'$, and $\zeta_i'$ are the Riccati–Bessel functions and their derivatives, which can be computed by the Bessel and Neumann function. According to Equations (3) and (4), when the incident wavelength and refractive index are fixed, the particle size of coal dust mainly affects the laser attenuation characteristics. In practice, the incident wavelength is a parameter of the rangefinder, and the refractive index of coal dust is almost fixed, so both can be regarded as constant values. Only the particle size of coal dust varies, so it significantly affects the laser attenuation and ranging error.

The laser propagation in a coal dust environment is shown in Figure 1. It is assumed that the incident power at a distance $x$ from the light source $O$ is $W_{\mathrm{inc}}(x)$, the effective cross-sectional area is $S(x)$, and the number density of particles is $\rho(x)$. Further, considering the single scattering of multiple particles, the output power $W_{\mathrm{out}}(x)$ at the position $x$ is

$$W_{\mathrm{out}}(x) = W_{\mathrm{inc}}(x) - C_{ext}I_{\mathrm{inc}}(x)S(x)\rho(x)\mathrm{d}x, \tag{5}$$

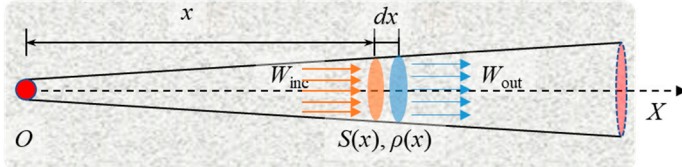

**Figure 1.** Laser attenuation model under single scattering of multiple particles.

In Equation (5), $I_{\mathrm{inc}}(x)$ denotes the power density of the incident laser. According to the definition of power density, Equation (5) can be differentiated as follows:

$$-\mathrm{d}W_{\mathrm{inc}} = C_{ext}W_{\mathrm{inc}}\rho(x)\mathrm{d}x, \tag{6}$$

The incident power at the position $x$ can be obtained by integrating both sides of Equation (6):

$$W_{\text{inc}}(x) = W_0 \exp\left(-\int_0^x C_{\text{ext}} W_{\text{inc}} \rho(x) \mathrm{d}x\right), \tag{7}$$

Assuming that the particle size in the laser propagation path follows a certain distribution $f(r)$, the expected particle number density $\rho$ can be estimated as follows:

$$\mathrm{E}(\rho(x)) = \int_{-\infty}^{\infty} \frac{m(x)}{V(r^3)\sigma} f(r) \mathrm{d}r, \tag{8}$$

where $V$ is the volume function, and $m(x)$ is the coal dust mass concentration. According to Equation (8), the particle concentration is also a key factor affecting the laser attenuation besides the particle size. Therefore, this study focuses on the influence of coal dust particle size and concentration on power attenuation and ranging error.

*2.2. Theory of Phase Ranging*

Among various ranging principles, phase ranging is one of the most precise techniques and is widely applied in current laser rangefinders as a primary ranging principle. The method calculates the distance by measuring the round-trip phase difference between the laser light source and the reflecting target. When the distance between the light source and target is $x$, and the electromagnetic wave velocity is $c$, the round-trip time $t$ is

$$t = \frac{2x}{c}, \tag{9}$$

when the laser modulation wavelength is $\lambda$, the round-trip phase difference can be obtained as:

$$\Delta\Phi = 2\pi\frac{c}{\lambda}t = 2\pi(N + \Delta N), \tag{10}$$

where $N$ is the integral part of the number of cycles and $\Delta N$ is the fractional part. Substituting Equation (10) into Equation (9), the relationship between distance and phase difference can be derived as follows:

$$x = \frac{\lambda}{2}(N + \Delta N), \tag{11}$$

where $\frac{\lambda}{2}$ is called the optical ruler. It can be seen that the ranging result $x$ consists of both a full ruler $N \times \frac{\lambda}{2}$ and residual ruler $\Delta N \times \frac{\lambda}{2}$. Because the phase value ranges between $[0, 2\pi)$, only the residual phase can be measured, and the number of full cycles cannot be obtained. Therefore, the rangefinders modulate lasers with different wavelengths to form a group of optical rulers. The number of full cycles measured by the longest optical ruler is 0, and its residual cycle can obtain a rough ranging result (limited by the phase difference measuring accuracy). The shorter optical rulers only calculate the residual length to compensate for the accuracy of rough ranging results. Therefore, for a group of optical rulers, the phase ranging result is

$$D = \frac{c}{2}\sum_i \frac{\Delta\varphi_i}{2\pi f_i}, \tag{12}$$

where $\Delta\varphi_i$ denotes the offset of phase measured by the $i$th optical ruler and $f_i$ denotes the modulation frequency of the $i$th optical ruler. For a rangefinder, the group of optical rulers is fixed, so any change in the light speed can directly affect the ranging results. According to Maxwell's electromagnetic theory, the relationship between the speed of light, wavelength, and medium refractive index is

$$n = \frac{c_0}{c} = \frac{\lambda_0}{\lambda}, \tag{13}$$

where $n$ denotes the refractive index of the medium. $c_0$ and $\lambda_0$ are the speed and wavelength of light in a vacuum, respectively, while $c$ and $\lambda$ denote their counterparts in the medium.

When the refractive index in the environment changes, the phase ranging error can be expressed as follows:

$$\delta d = D - D' = \left(\frac{1}{n} - \frac{1}{n'}\right) c_0 \sum_i \frac{\Delta \varphi_i}{4\pi f_i} = \left(\frac{n'}{n} - 1\right) D', \tag{14}$$

where $D$ and $D'$ are the true and measured values of distance, respectively, while $n$ and $n'$ are the true and theoretical values of the environmental refractive index, respectively. Substituting Equation (13) into Equation (14), the relationship between ranging error and relative change in laser wavelength can be established. Therefore, the ranging error can be estimated by measuring the relative change in the laser wavelength in the coal dust environment, which can be used to correct the measured distance as follows:

$$\delta d = \left(\frac{\lambda}{\lambda'} - 1\right) D', \tag{15}$$

## 3. Experimental Data

This study aims to investigate the impact of particle size and coal dust mass concentration on laser power attenuation and wavelength change. Mathematical models that use experimental results are established to describe these effects.

### 3.1. Experiment Setting

The experiment setup is shown in Figure 2, which consists of an optical detection module, a diffusion module, and a concentration measurement module.

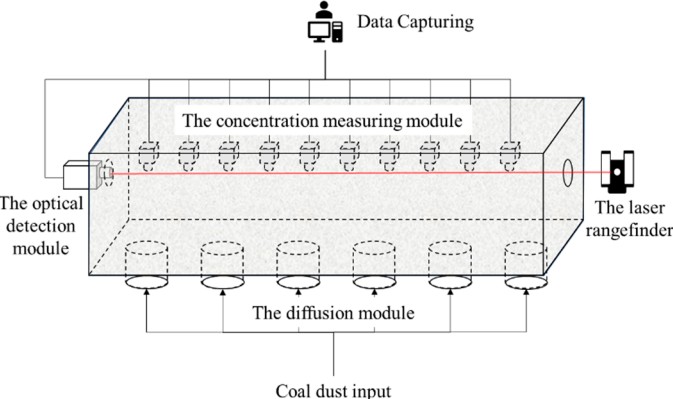

**Figure 2.** Schematic of the experimental facility for coal dust analysis.

The optical detection module measures the laser power and wavelength in both dust-free and dusty environments, while the diffusion module attains steady-state dust diffusion for each particle size. Additionally, the concentration measurement module measures dust mass concentration at multiple points along the laser propagation path.

Figures 3 and 4 depict the experimental components. A total station serves as the laser transmitter and rangefinder, and an optical detector measures the laser power and wavelength. A prism acts as a cooperative target for the rangefinder. The diffusion module comprises an acrylic box, a blast blower, and several pipes. The concentration measurement module consists of a precision balance, a dust concentration meter, and a series of sampling position labels.

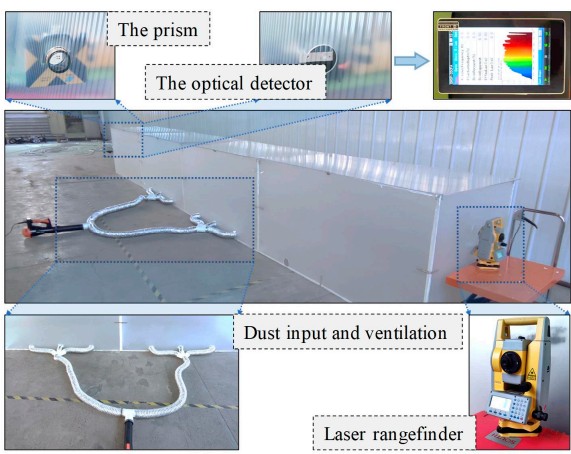

**Figure 3.** Components of experimental facility.

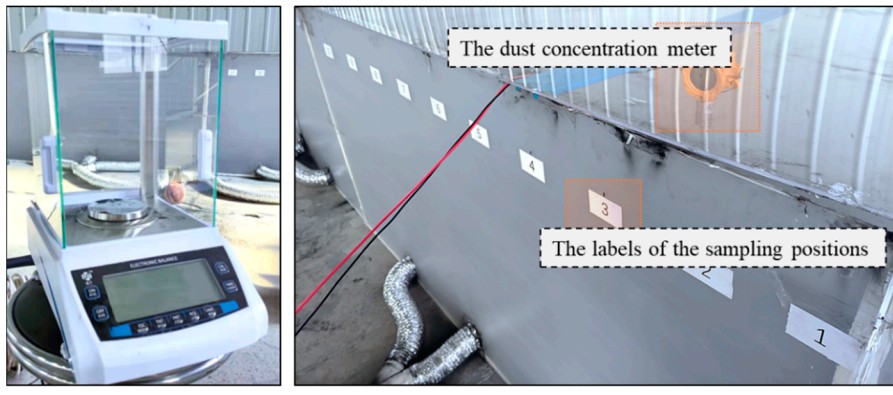

(a) The precise balance      (b) The concentration measuring on the propagation path

**Figure 4.** Control and measurement of coal dust concentration.

*3.2. Materials*

The coal dust material was separated into groups based on their particle size levels. To determine the division intervals, it was necessary to obtain the particle size distribution by a granulometer. According to the test, the size distribution of the coal dust sample is shown in Figure 5.

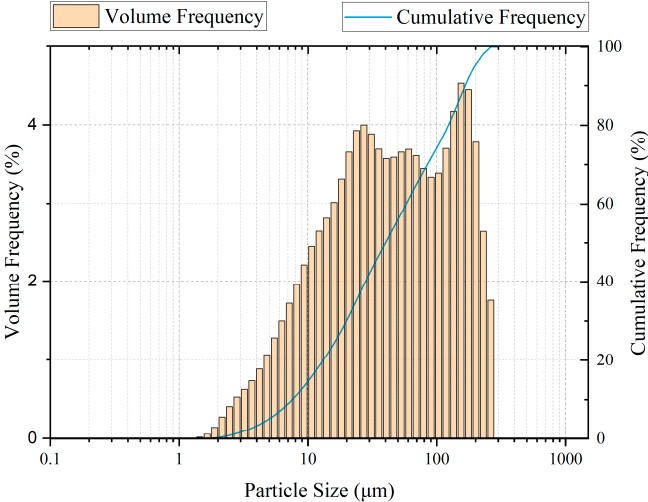

**Figure 5.** Particle size distribution of the coal dust sample used in this study.

Figure 5 reveals the non-uniform distribution of the coal dust sample, which mainly ranges from 10 to 230 μm, with an average size of 65.631 μm. Based on the particle size distribution, the number and scales of sieves were selected according to available conditions. The link between the scale of sieves and the corresponding particle size is shown in Table 1.

**Table 1.** Scales of the selected sieves and the corresponding particle size.

| Scale | 600 | 325 | 230 | 170 | 120 | 100 | 80 | 70 | 60 |
|---|---|---|---|---|---|---|---|---|---|
| Size (μm) | <20 | [20, 45] | [45, 63] | [63, 90] | [90, 125] | [125, 150] | [150, 180] | [180, 212] | [212, 250] |

After sorting the material into various particle sizes, it was weighed using a precise balance and placed into the experimental setup. The coal dust concentration along the propagation path was measured using the diffusion module to establish the corresponding relationship between the input mass and the coal dust concentration.

The statistical analysis results for each sampling point are shown in Figure 6. It can be seen that in the same particle size group, the bias value (i.e., the difference between the measured and average values of the mass concentration) has an approximately normal distribution with 0 as the expectation value. In addition, the standard deviations of the measured values are less than 5.22 mg/m$^3$. Furthermore, it is clear from Figure 6 that in the same particle size group, the average dust mass concentration in the propagation path is almost linearly proportional to the input mass. Further, when the input mass remains the same, the larger the particle size, the lower the average concentration.

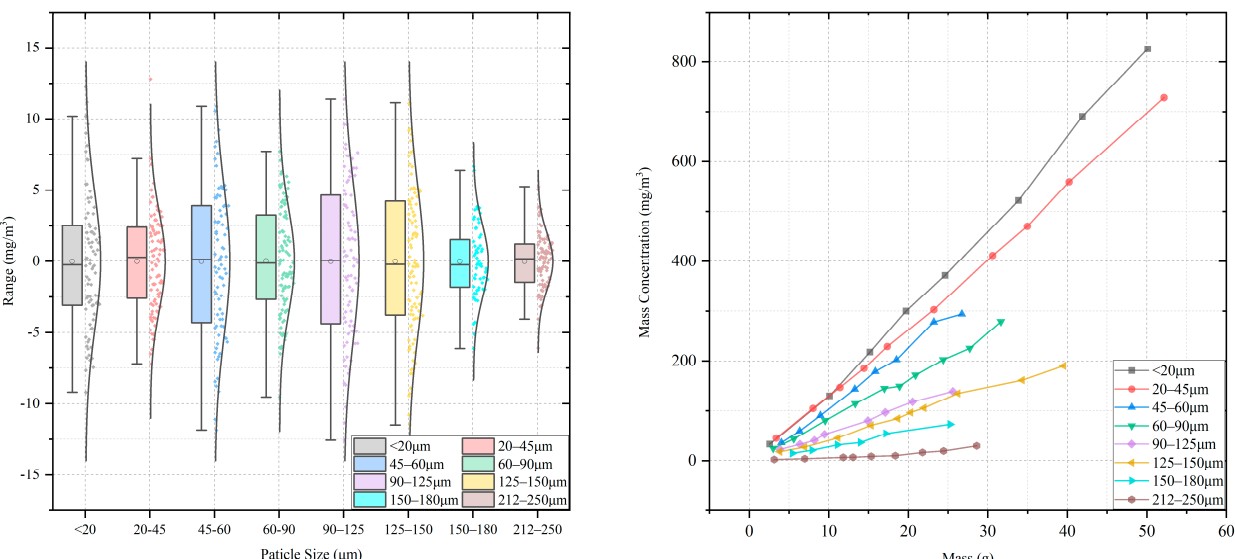

**Figure 6.** Relationship between the concentration and input mass.

## 4. Results and Discussion

### 4.1. Relationship between the Dust Mass Concentration and Power Attenuation

Due to the extinction effect of coal dust particles on electromagnetic waves, the intensity of the laser signal is attenuated when it reaches the receiver through the coal dust environment. After the coal dust material was divided into groups based on the particle size, experiments were conducted under varying masses, and the corresponding power attenuation ratio data were obtained. The results are shown in Figure 7.

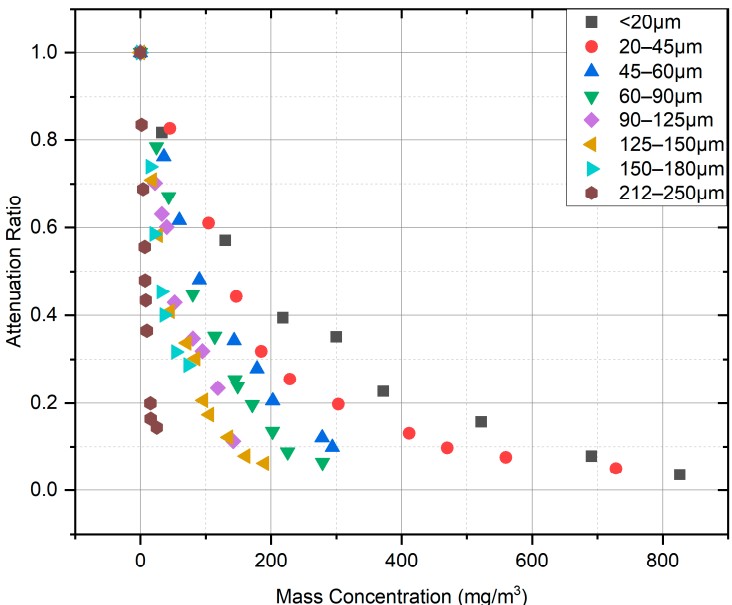

**Figure 7.** Relationship between the dust mass concentration and attenuation ratio for each particle size level.

It can be seen in Figure 7 that at the same particle size level, the increase in the mass concentration aggravates the power attenuation, but the attenuation speed gradually decreases with the increase in the mass concentration. The obtained plots were fitted with exponential attenuation ($y = y_0 + A \exp(Rx)$) and quadratic attenuation ($y = \left(a + bx + cx^2\right)^{-1}$) functions, and the results are shown in Figure 8 and Table 2.

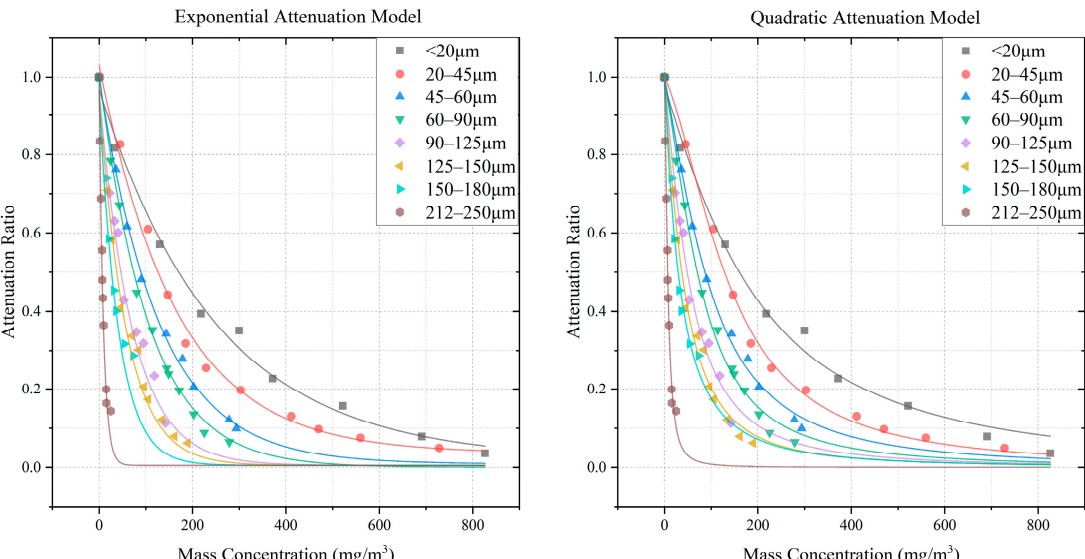

**Figure 8.** Fitting curves obtained by the exponential and quadratic attenuation functions.

It is clear from Figure 8 that under the same mass concentration, the larger the particle size, the more intense the attenuation. This may be attributed to the fact that the extinction cross section of coal dust particles with larger size is larger. To verify these results, Equation (10) is used to calculate the extinction cross section and extinction coefficient of the coal dust particles with sizes ranging from 1 μm to 200 μm.

**Table 2.** Fitting parameters of the exponential and quadratic attenuation models.

| Model | Param | <20 μm | 20–45 μm | 45–60 μm | 60–90 μm | 90–125 μm | 125–150 μm | 150–180 μm | 212–250 μm |
|---|---|---|---|---|---|---|---|---|---|
| exp | $y_0$ | 0.0050 | 0.0050 | 0.0050 | 0 | 0.0288 | 0.0200 | 0.0100 | 0.0050 |
|  | $A$ | 0.9471 | 0.9935 | 0.9874 | 1.0078 | 0.9578 | 0.9478 | 0.9373 | 0.9799 |
|  | $R$ | −0.0040 | −0.0060 | −0.0078 | −0.0095 | −0.0142 | −0.0171 | −0.0243 | −0.1085 |
| qua | $a$ | 1.0228 | 0.9998 | 1.0024 | 1.0096 | 1.0048 | 1.0027 | 0.9897 | 1.0102 |
|  | $b$ | 0.0042 | 0.0027 | 0.0066 | 0.0069 | 0.0136 | 0.0194 | 0.0292 | 0.0639 |
|  | $c$ | $1.1883 \times 10^{-5}$ | $3.9714 \times 10^{-5}$ | $5.6893 \times 10^{-5}$ | $1.0274 \times 10^{-5}$ | $1.4072 \times 10^{-4}$ | $1.9865 \times 10^{-4}$ | $1.7824 \times 10^{-4}$ | 0.0115 |

Figure 9 shows that with the increase in particle size, the extinction coefficient of coal dust gradually decreases while the extinction cross section increases. This provides a qualitative explanation for the results in Figure 8 and is consistent with the relationship between the extinction section and attenuation ratio in Equation (14). In addition, Table 2 shows that in the exponential attenuation model, with the increase in particle size, the power parameter $R$ decreases, and the intercept $y_0$ and coefficient $A$ do not show any specific variation trend. In the quadratic attenuation model, the first coefficient, $b$, grows with the increase in the particle size, while $a$ and $c$ do not show any variation trend. To select the model with better fitness, their residuals are calculated, as shown in Figure 10.

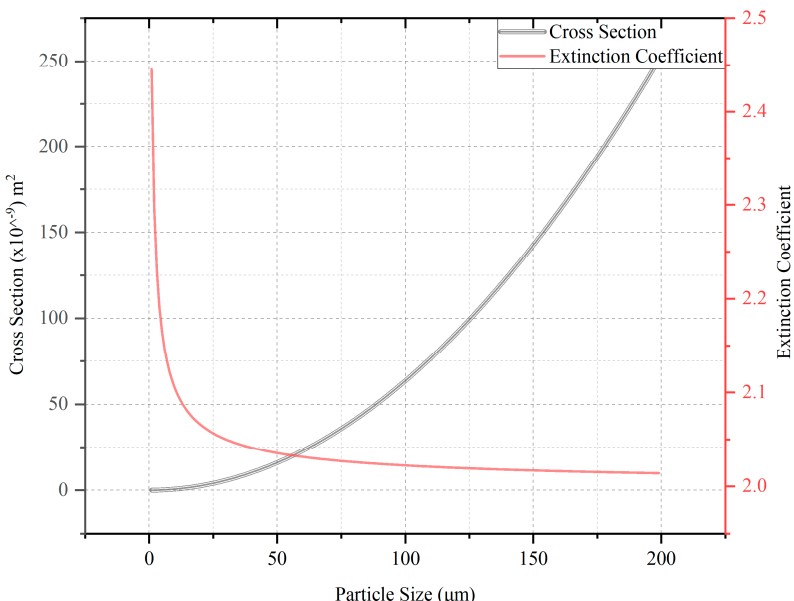

**Figure 9.** Extinction cross section and extinction coefficient computed based on Mie theory.

It is evident from Figure 10 that under the same particle size level, the error of the box graph obtained by the exponential attenuation model is less dispersed, and the expected value is closer to 0 than that obtained by the quadratic attenuation model, indicating that the exponential attenuation model has higher precision and accuracy. Therefore, the exponential attenuation model is more effective for establishing the relationship between the coal dust mass concentration and laser power attenuation.

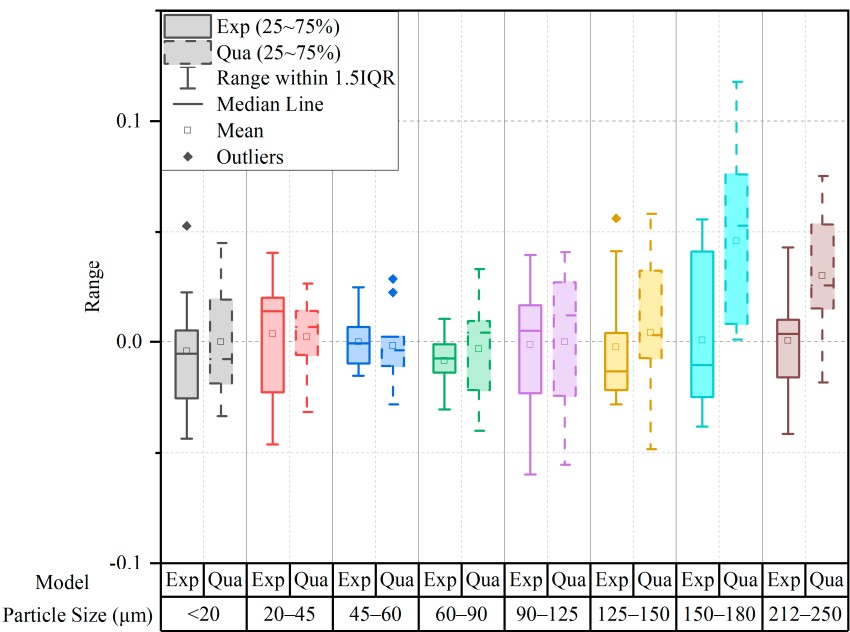

**Figure 10.** Residual statistics of exponential attenuation and quadratic attenuation models.

*4.2. Estimation of the Effective Distance for the Laser Rangefinder in a Coal Dust Environment*

During the experiments, we found that the rangefinder fails when the laser power attenuates below a certain threshold value at the receiving end. After repeated experiments, the expected value of the power threshold was estimated to be 6.65 $\mu W/cm^2$, and the expected value of the transmitted power was 107.25 $\mu W/cm^2$. According to the residual analysis in Section 4.1, the exponential attenuation model with better fitting performance is selected to estimate the cut-off distance of the rangefinder.

According to Lambert–Beer law, the power at the receiver end is related to the attenuation coefficient and propagation distance, i.e.,

$$\frac{W(x)}{W_0} = \exp(-k_\lambda x), \tag{16}$$

where $k_\lambda$ denotes the attenuation coefficient of the medium. Because the length of the propagation path is fixed in the experiment, the attenuation coefficient has an exponential relationship with the power ratio. Combined with the exponential relationship between the mass concentration and power ratio, it can be inferred that there exists a linear relationship between the coal dust mass concentration and the attenuation coefficient (i.e., $k_\lambda = a_0 + a_1 \times m$). Based on this assumption, model fitting is carried out. The fitting effect and related parameters are shown in Figure 11 and Table 3, respectively.

**Table 3.** Parameters of the linear fitting model between the concentration and attenuation coefficient.

| Param | <20 μm | 20–45 μm | 45–60 μm | 60–90 μm | 90–125 μm | 125–150 μm | 150–180 μm | 212–250 μm |
|---|---|---|---|---|---|---|---|---|
| $a_0$ | 0.0061 | 0.0445 | 0 | −0.0088 | −0.0002 | 0.0246 | 0.0303 | 0.0237 |
| $a_1$ $(\times 10^{-4})$ | 9.4846 | 10.7 | 19.2 | 25.2 | 34.3 | 36.8 | 44.4 | 216.6 |

As shown in Figure 11, the mass concentration is linearly proportional to the attenuation coefficient. At the same particle size level, the attenuation coefficient increases as the mass concentration increases. At the same mass concentration, the greater the particle size, the larger the attenuation coefficient. It can be seen in Table 3 that the coefficient $a_1$ increases with the increase in particle size, while the intercept $a_0$ does not show any specific variation trend.

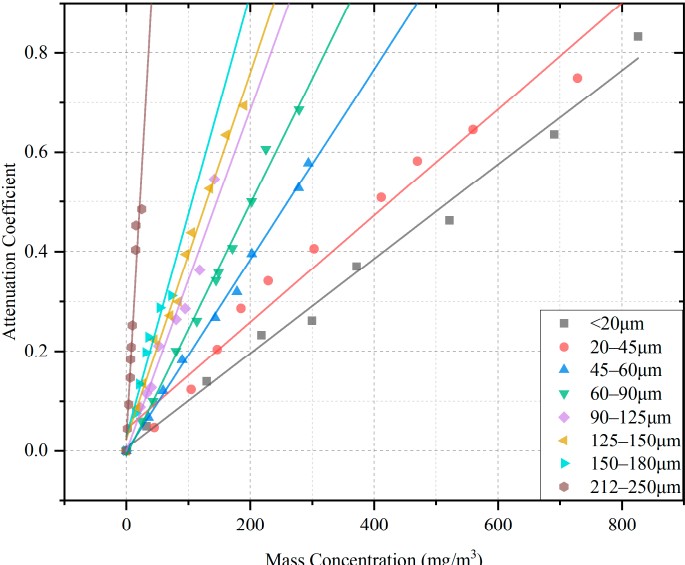

**Figure 11.** Linear fitting between the mass concentration and attenuation coefficient.

According to the Lambert–Beer law, as well as the results in Figure 11 and Table 3, the cut-off distance of the rangefinder can be estimated as:

$$x = -\frac{\ln(\gamma_0)}{a_0 + a_1 \times m},\tag{17}$$

where $\gamma_0$ represents the attenuation ratio once the rangefinder fails. Through repeated tests, its value is found to be 0.0620. According to Table 3, the relationship between the coal dust mass concentration and cut-off distance can be established for each particle size level, which is shown in Figure 12.

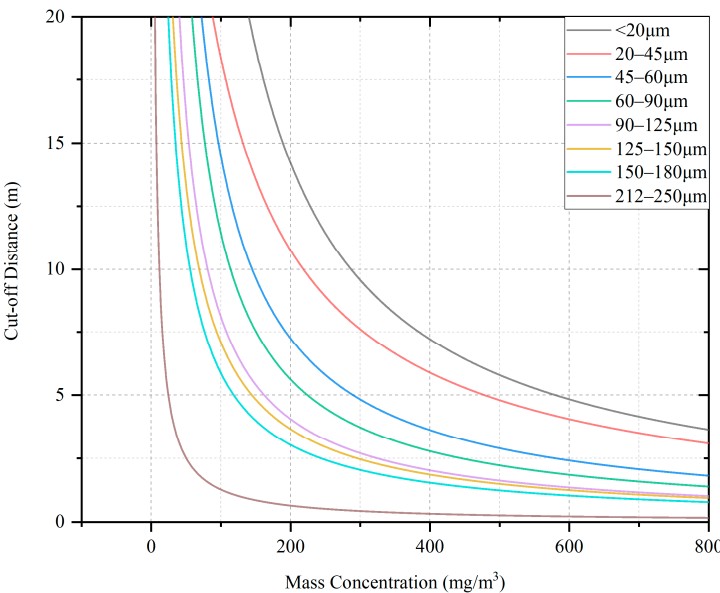

**Figure 12.** Estimated relationship between the mass concentration and cut-off distance.

Figure 12 shows that at the same particle size level, the cut-off distance rapidly decreases with the increase in mass concentration initially. At the same mass concentration, the smaller the particle size, the greater the cut-off distance. In practical applications, the cut-off distance can be estimated according to the selected particle size, mass concentration, and cut-off attenuation ratio to obtain the effective range of the instrument.

### 4.3. *Variation in the Laser Wavelength and Ranging Correction in a Coal Dust Environment*

With the laser wavelength measured in a dust-free environment as the benchmark, the relative variations in the laser wavelength under different coal dust particle sizes and mass concentrations were determined. The relative refractive index was calculated using Equation (13). The results are shown in Figure 13.

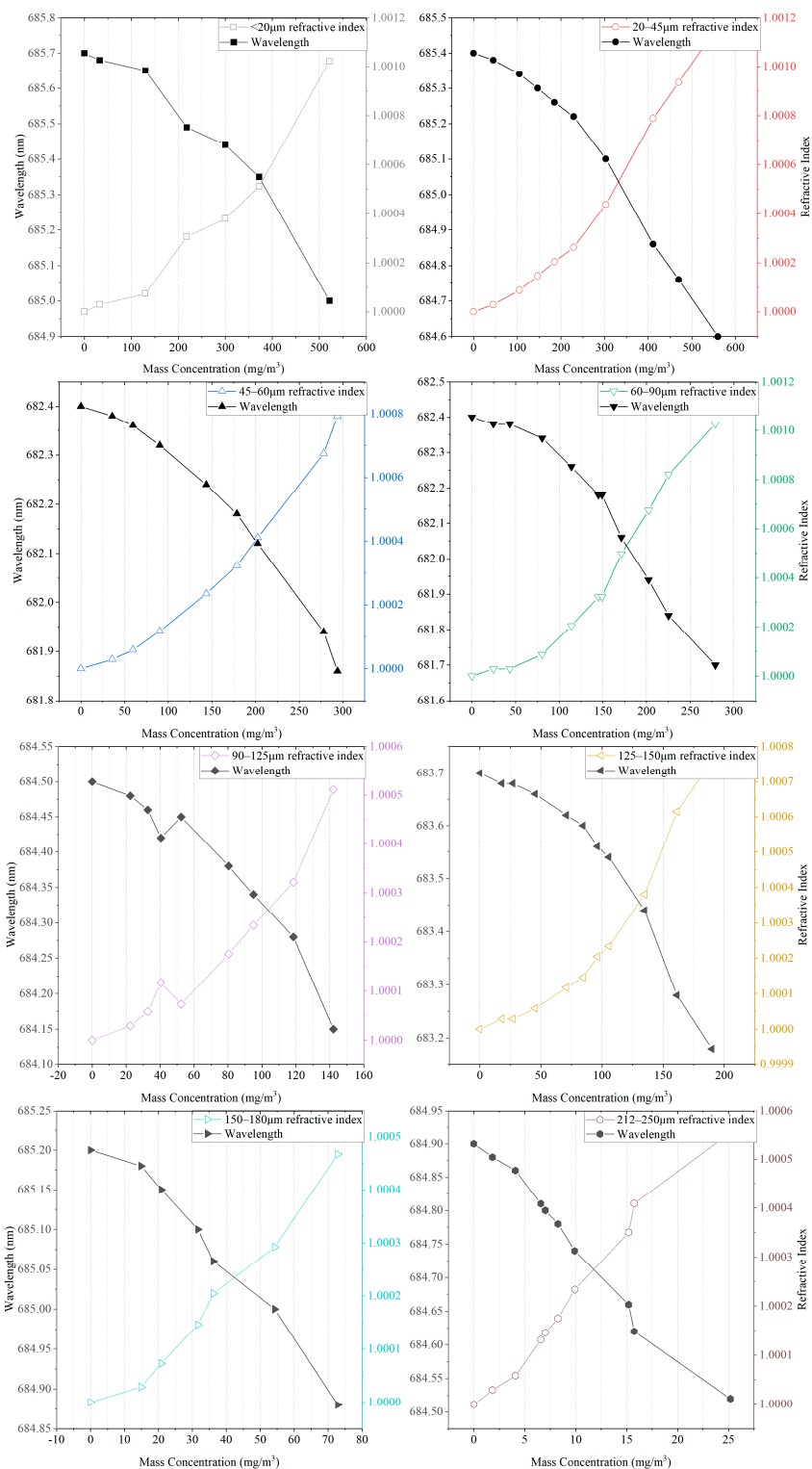

**Figure 13.** Variation in the wavelength and refractive index at each particle size level.

Figure 13 shows that with the increase in the coal dust mass concentration, the laser wavelength decreases, while the relative refractive index increases, and their growth rates increase with the increase in coal dust mass concentration. A polynomial model was used to fit the obtained curves at each particle size level. The results and relevant parameters are shown in Figure 14 and Table 4, respectively.

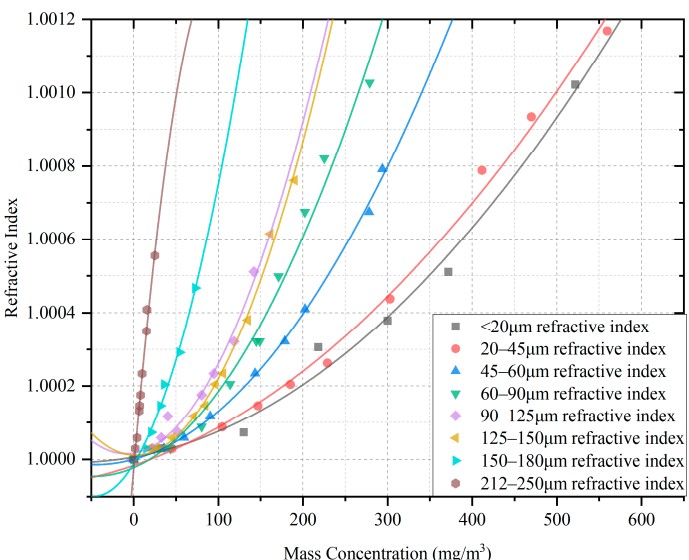

**Figure 14.** Fitting curves for the relationship between mass concentration and refractive index.

**Table 4.** Parameters of the fitting model for concentration vs. refractive index curve ($y = b_0 + b_1 m + b_2 m^2$).

| Param | <20 μm | 20–45 μm | 45–60 μm | 60–90 μm | 90–125 μm | 125–150 μm | 150–180 μm | 212–250 μm |
|---|---|---|---|---|---|---|---|---|
| $b_0$ | 1.00001 | 0.99998 | 1 | 0.99998 | 1.00002 | 1.00001 | 0.99999 | 0.99998 |
| $b_1\ (\times 10^{-7})$ | 4.0844 | 7.811 | 6.2578 | 9.9547 | 3.631 | −92.7828 | 37.077 | 268.202 |
| $b_2\ (\times 10^{-7})$ | 0.02889 | 0.02517 | 0.06795 | 0.10753 | 0.20799 | 21.8649 | 0.03921 | −1.3084 |

As shown in Figure 14, with the increase in particle size, the growth of the relative refractive index becomes more rapid. This change can be quantitatively expressed using the data in Table 4. In addition, an increase in the refractive index implies a decrease in the laser propagation speed, which can introduce errors into the rangefinder, making the measured value smaller than the actual distance. According to Equation (7), the range correction value can be calculated, and its comparison with the true value is shown in Figure 15.

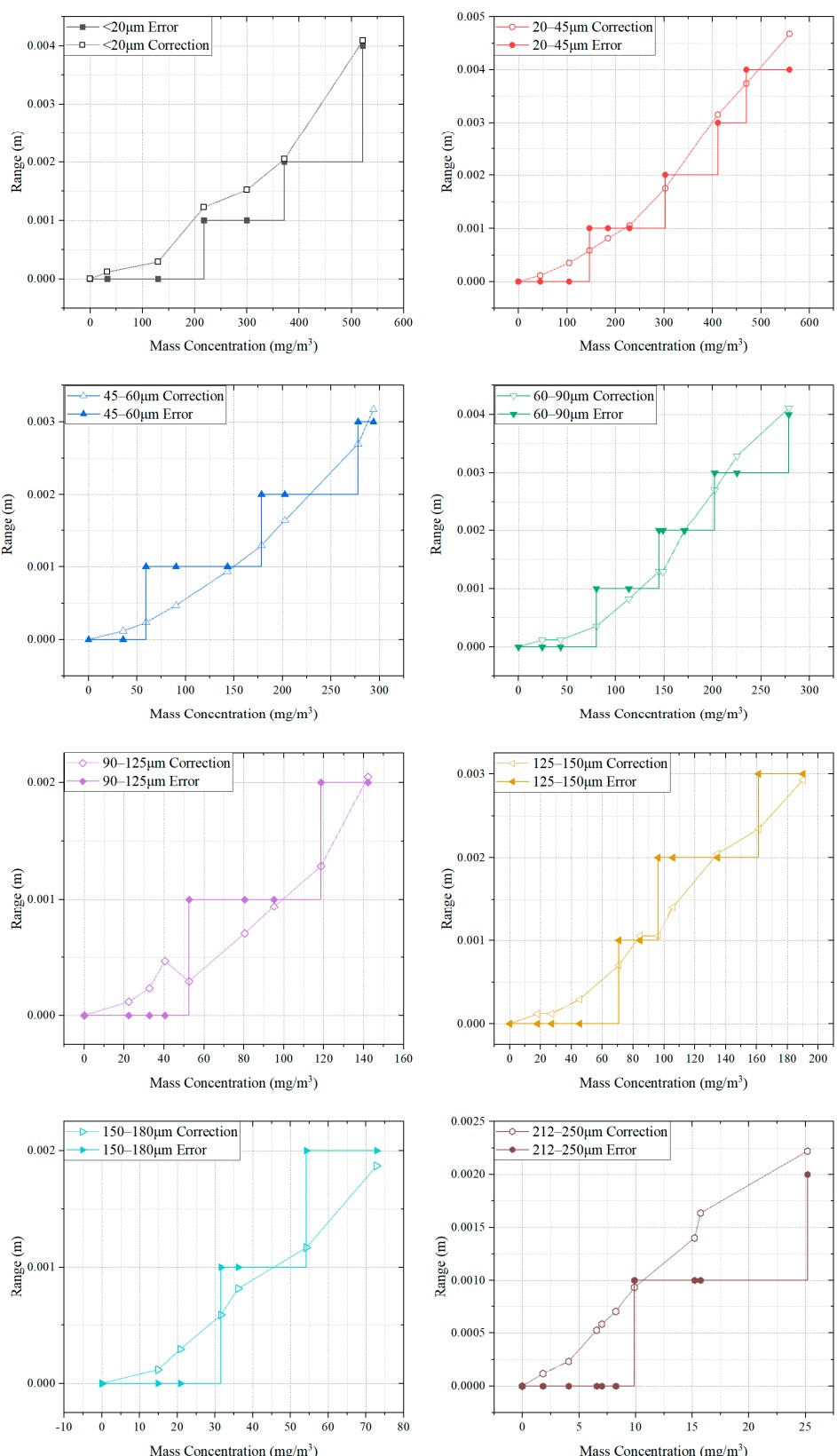

**Figure 15.** Errors and correction values at each particle size level.

Figure 15 shows that both true error and theoretical correction values are positively correlated with the coal dust mass concentration. Since the resolution of the rangefinder is 1 mm, the true error shows sudden jumps, which are shown by the horizontal stepped

lines. The theoretical correction value is continuously calculated according to the model, so the line diagrams are used to show its variation trend. It can be seen in Figure 15 that there exist differences between the theoretical correction values and the true errors. In addition, in many positions, the theoretical correction value is less than the true error, which may be due to the intrinsic error of the correction model and the limited wavelength measurement accuracy. Further, the true error itself has a precision of only a few millimeters. The root mean square error (RMSE) is used to calculate the average accuracy as follows:

$$\xi_r = \sqrt{\frac{\sum(\delta d - \Delta)^2}{n}},$$ (18)

where $\xi_r$ denotes the RMSE at the particle size level $r$, $\Delta$ denotes the true error, and $\delta d$ denotes the theoretical correction value, which is calculated by Equation (6). According to Equation (18), the average RMSE results for each particle size level are obtained, as shown in Table 5 and Figure 16.

**Table 5.** Average RMSE of ranging correction method at each particle size level.

| Size Level (μm) | <20 | 20–45 | 45–60 | 60–90 | 90–125 | 125–150 | 150–180 | 212–250 |
|---|---|---|---|---|---|---|---|---|
| RMSE (m) | $2.7 \times 10^{-4}$ | $3.1 \times 10^{-4}$ | $4.52 \times 10^{-4}$ | $4.09 \times 10^{-4}$ | $4.2 \times 10^{-4}$ | $4.700 \times 10^{-4}$ | $4.100 \times 10^{-4}$ | $4.467 \times 10^{-4}$ |

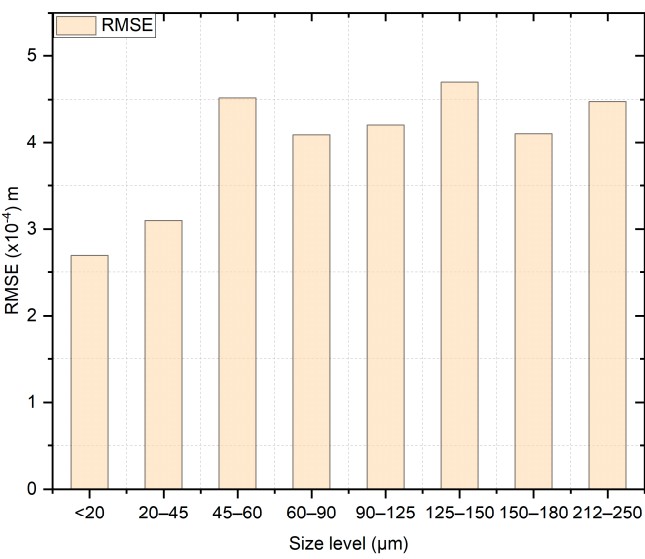

**Figure 16.** Average RMSE of ranging correction method at each particle size level.

Table 5 and Figure 16 shows that the RMSE of this correction method ranges from 0.27 to 0.47 mm, which indicates a better precision than that of the laser rangefinder. This verifies the efficacy and feasibility of the proposed correction model.

## 5. Conclusions

To address the challenges of laser power attenuation and inaccurate laser ranging in coal dust environments, we established power attenuation and ranging correction models based on theoretical analysis and experimental verification. The particle sizes and mass concentrations of the coal dust were considered as the two main influencing factors. In summary, this research work can be concluded as follows:

Based on Mie theory and the multi-particle single-scattering model, the laser power attenuation in a coal dust environment was examined. At the same particle size level, with the increase in coal dust mass concentration, the power drops, but its speed gradually slows down. The exponential attenuation model shows better fitting performance than the quadratic attenuation model. At the same mass concentration, coal dust with a larger

particle size has a stronger influence on laser attenuation than smaller sizes. This is in accord with Mie theory: the larger the particle size, the greater the extinction cross section, which can lead to a stronger attenuation in the single scattering model.

Based on the proposed exponential attenuation model and Lambert–Beer law, the laser attenuation coefficient at each particle size level is estimated, and the corresponding relationship between the mass concentration and ranging cut-off distance is calculated. In practical applications, the cut-off distance can be used to obtain the effective range of the rangefinder.

Based on the phase ranging principle and Maxwell's electromagnetic theory, a ranging correction method is proposed to determine the laser wavelength at different particle size levels. A polynomial model is used to fit the relative refractive index at each particle size level. The correction efficiency is evaluated by comparing the theoretical correction value with the true error. The results show that the RMSE of the correction values varies from 0.27 to 0.47 mm, which verifies the effectiveness of the proposed correction model.

Future research can utilize the proposed attenuation model and correction method in actual coal mine working faces to suppress laser rangefinder errors in coal dust environments to improve the quality of surveying and mapping and provide high-precision data support for the construction of intelligent coal mines. Limited by our investigating level and resource, the grain of the experimental material could not be finer. Moreover, the precision of the used laser rangefinder may affect the accuracy of the proposed model. In the future, the other underlying factors of laser attenuation and ranging errors can be investigated with better experimental conditions.

**Author Contributions:** Conceptualization, S.M. and B.L.; methodology, B.L.; software, B.L.; validation, B.L. and H.Z.; formal analysis, S.M. and B.L.; investigation, B.L.; resources, S.M.; data curation, B.L.; writing—original draft preparation, B.L.; writing—review and editing, S.M.; visualization, B.L.; supervision, S.M.; project administration, S.M.; funding acquisition, S.M. All authors have read and agreed to the published version of the manuscript.

**Funding:** This research was funded by the National Key R&D Program of China, grant number 2020YFB1314001.

**Institutional Review Board Statement:** Not applicable.

**Informed Consent Statement:** Not applicable.

**Data Availability Statement:** The data are contained within the article. The data presented in this study are available at https://figshare.com/articles/dataset/Experiment_origin_data_xlsx/22229890 (accessed on 8 March 2023).

**Acknowledgments:** The authors want to thank the engineers of Beijing Longruan Technology Co., Ltd. for the experiment environment establishment.

**Conflicts of Interest:** The authors declare no conflict of interest.

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
