# Peer review of "Laser Attenuation and Ranging Correction in the Coal Dust Environment Based on Mie Theory and Phase Ranging Principle"

_atmosphere, doi:10.3390/atmos14050845_

Round 1

Reviewer 1 Report

In this paper the problem of laser ranging in coal mine dust environment was studied, which does not belong to the research scope of this journal. It is suggested that this article should be transferred to other related journals. In addition, there are the following problems: All variables in all formulas have not been assigned their measurement units. In Figure 5, the abscissa in the subfigure on the left lacks a coordinate caption, and the ordinate lacks a unit.

Author Response

Thank you for commenting and giving conductive suggestions about our research.

(1)Our primary objective is to improve the feasibility of laser facilities in coalmines, ultimately leading to labor savings. And in the paper, we adopted methods of modeling and simulation. For these reasons, we submitted our manuscript to the special issue “New Insights into Human Health by Air Quality Modeling, Simulation, and Observation”. Upon your suggestion, we have added emphasis on the significance of our work to human health in the Abstract, Introduction, and Conclusion sections.

(2)As for the problem of variables unit in formulas, we added a statement that unless specified, the units are SI Units in default.

(3)And we added units and captions to coordinate axes of all figures.

Reviewer 2 Report

Comments and Suggestions for improvement:

1. The original contributions need to be much better presented in the last paragraph of the section “INTRODUCTION”. All improvements, if they are, and new results must be described in this paragraph. The advantages of the work are not discussed in the text. Indeed, the introduction needs to enrich the readers with state-of-the-art works, which gives the reader a clear vision of the gap that those studies have not addressed and covered in this study.

2. A brief description of the structure (layout) of the paper may be added to the end of the Introduction.

3. The Abstract should modify by adding advantages of the proposed method.

4. Check the manuscript carefully for typos and grammatical errors.

5. Please clarify the novelty of this paper with respect to the published paper.

6. The references list is not at all updated with the latest developments and publications.

7. In general, the typeset equations should be regarded as parts of a sentence and treated accordingly with the appropriate grammatical convention and punctuation. More editing for writing is needed. At the end of all equations must be put in “COMMA” or “POINT” according to the typing rules.

8. All acronyms should be defined before.

9. Please rewrite the paper in the Template of the Journal. The references of this paper should be rewritten accordingly to the style of the journal.

10. Section Conclusion should be elaborated more in detail.

 11. The authors have presented some figures without any physical explanation besides the resolution of these figures is very bad. Try to increase the resolution and size of the labels of the figures.

Author Response

Thank you for commenting and giving conductive suggestions about our research. We have made the following corrections to our paper.

  1. In the Introduction Section, we have placed greater emphasis on the key elements of our research. Our study seeks to address the gaps in knowledge concerning coal dust environments due to the paucity of ranging error correction methods reported in prior studies, and the heavy reliance on empirical methods.
  2. We added a paragraph to describe the organization of this paper in Section Introduction.
  3. We have revised the Abstract to highlight the advantages of our research, such as the use of both theoretical and empirical approaches, the high accuracy of our proposed attenuation model, and the applicability of our ranging correction method. Additionally, we have included key findings in the Abstract to make it more quantitative.
  4. We have thoroughly checked the full text of the paper for errors and corrected any that were found.
  5. To highlight the novelty of our research, we have outlined the shortcomings of prior studies, which lack the combination of theoretical and empirical approaches, the high accuracy of our proposed attenuation model, and the applicability of our ranging correction method.
  6. We reviewed and cited relevant papers published in 2022 in our manuscript.
  7. We added commas at the end of every equation.
  8. We checked the full text and defined the missing acronyms at their first appearance.
  9. We followed the Template of the Journal to write this paper, and it has passed the check from the assistant editor.
  10. In the Conclusion Section, we have improved the logicality of our argument, highlighting the outcomes of our research by summarizing them following a sequential logic. Additionally, we have provided concrete form to our proposed models in order to elaborate the conclusions.
  11. We have added descriptions to the figures that lacked them, and substituted unclear figures with their high-resolution versions.

Reviewer 3 Report

In the study, a laser power attenuation model and a ranging correction model are proposed considering the dust particle size and mass concentration. Based on the conducted experiments, the proposed model and correction method were proved that can be effectively used in the coalmine environment to reduce the laser rangefinder error, improve the accuracy of underground measurements, and provide high-precision data reference for the construction of intelligent coalmines. It can be accepted after minor revision.

1.      It is suggested to added relevant reference in Line 76 if there are investigations on the laser attenuation models and ranging error correction methods.

2.      It is suggested to combine the Figure 6 into a single figure, which can better compare the effect of particle size.

3.      There still exists some errors: Line 305, there is a useless blank, check the whole manuscript.

4.      Can the authors transfer the Table 5 into a figure to improve its intuitiveness.

5.      It seems that there is no Figure 9 in the manuscript. Check about it. In addition, it is suggested to add legend about the difference between solid and dotted boxes.

Author Response

Thank you for commenting and giving conductive suggestions about our research. Our corrections are as follows.

  1. We have introduced the relevant investigations on laser attenuation and ranging correction methods before line 74. Specifically, the formerly mentioned methods were classified into Mie-based and simulation-based methods, while the latter were evaluated for errors in particle environment. However, to our knowledge, no appropriate method for correcting the ranging errors in a coal dust environment has been proposed.
  2. We merged the subplots of Figure 7 into a single one.
  3. We have thoroughly checked the full text for typos or other written errors.
  4. We have added Figure 15 as an illustration of Table 5.
  5. We have remedied the error of the missing Figure 10. Regarding the legend of this figure, the solid boxes and the dashed boxes correspond to the exponential and quadratic models, respectively, and this has been added as a legend for the figure.

Reviewer 4 Report

This subject addressed is within the scope of the journal. However, the manuscript in the present version contains several problems. Appropriate revisions should be undertaken in order to justify recommendation for publication.

1. It is mentioned that
Mie Theory and Phase Ranging Principle are used. What are the advantages of adopting these particular methods over others in this case? How will this affect the results? More details should be furnished.

2.      For readers to quickly catch your contribution, it would be better to highlight major difficulties and challenges, and your original achievements to overcome them, in a clearer way in abstract and introduction.

3. There is a serious concern regarding the novelty of this work. What new has been proposed?

4. Abstract needs to modify and to be revised to be quantitative. You can absorb readers' consideration by having some numerical results in this section.

5. There are some occasional grammatical problems within the text. It may need the attention of someone fluent in English language to enhance the readability.

6. The discussion section in the present form is relatively weak and should be strengthened with more details and justifications.

7. In conclusion section, limitations and recommendations of this research should be highlighted.

8. The authors have to add the state-of-the art references in the manuscripts.

Author Response

Thank you for commenting and giving conductive suggestions about our research. Our corrections are as follows.

  1. The particle sizes of coal dust in mining face are typically varying from 10 μm to 300 μm. Mie theory is appropriate to any particle size while other methods usually require assumptions to the particle size. As a result, we selected Mie theory as our researching basis. Among many ranging principles, the Phase Ranging is one of the most accurate techniques and is widely used in current laser rangefinder as the ranging principle. Therefore, we selected this principle as the analyzing basis in order to keep the novelty of our research. We added above statements in the manuscript to highlight the advantages of our theoretical basis.
  2. We described the research gaps including the lack of attention on coal mine environment, the heavy dependency on empirical methods and the insufficient compensation between theories and experiments. Our research went from solid theoretical foundation and extracted basic factors of experiments. As for the outcomes, we proposed practical attenuation model and ranging correcting model. We improved our expression in the Abstract and the Introduction.
  3. Our research's novelty lies in the proposed ranging correction model for coal dust environments, which are yet to be thoroughly investigated and applied. Previous studies neglected deep influencing factors and migration. Instead, we extracted two key factors, particle size, and mass concentration, based on Mie theory. These factors served as the basis for our experiments and the proposed ranging correction method. Our model's intuitive nature facilitates its transferability. We provided explanations of the results corresponding to the related Mie theory and multi-particle single-scattering model in the discussion section.
  4. We added some meaningful results data of the experiment into Abstract to make it more quantitative.
  5. We meticulously revised the manuscript to correct grammatical errors and typos.
  6. We improved the discussion on some experiment results in the manuscript. For instance, we merged the subplots of Figure 7 to a single one to form intuitively contrast. And we added Figure 16 to illustrate the RMSE of ranging correction. We also explained the results corresponding to related Mie theory and multi-particle single-scattering model, in discussion part.
  7. In the Conclusion section, we provided prospects for further investigation, such as limiting the material grain and device precision. We suggested that future studies consider other underlying factors that affect attenuation and ranging errors and utilize both better experimental methods and more rigorous theories.
  8. We have reviewed some relevant literature published in 2022 and cited them in our manuscript.

Round 2

Reviewer 1 Report

In this paper, the precise measurement of coal dust in the environment of coal dust by laser dust detector is studied. The research results have certain guiding significance for coal dust prevention and control. There are some minor problems with this article:

1. The selected Keywords do not meet the requirements. Suggested keywords are: dust control; coal dust control; dust measurement; dust ranging correction; dust laser-measurement.

2. The conclusion should be described in text. The formula should not appear in the conclusion.

Author Response

We are very appreciated that you reviewed and gave us suggestions on our manuscript again. We believe these suggestions conductive to improve the quality of the article. Accordingly, we have made some modifications to our manuscript based on your recommendations.

  1. In order to ensure that our article is more relevant to the special issue, "New Insights into Human Health by Air Quality Modeling, Simulation, and Observation", we have changed some of the keywords. Specifically, we have substituted "ranging correction" with "dust ranging correction", "coal dust environment" with "coal dust control", and "power attenuation" with "laser attenuation". Furthermore, we have reorganized these keywords in the following order: "coal dust control; laser attenuation; dust ranging correction; Mie theory; phase ranging".
  2. We have decided to delete the formula in the Conclusion section to standardize the manuscript.

Reviewer 4 Report

Accept in present form

Author Response

We are very appreciated that you reviewed and gave us suggestions on our manuscript.